# Immunomodulatory Effects of Macrolides Considering Evidence from Human and Veterinary Medicine

**DOI:** 10.3390/microorganisms10122438

**Published:** 2022-12-09

**Authors:** Joseph M. Blondeau

**Affiliations:** 1Department of Clinical Microbiology, Royal University Hospital and Saskatchewan Health Authority, Saskatoon, SK S7N 0W8, Canada; joseph.blondeau@saskhealthauthority.ca; 2Department of Microbiology and Immunology, Pathology and Laboratory Medicine and Ophthalmology, University of Saskatchewan, Saskatoon, SK S7N 0W8, Canada

**Keywords:** macrolides, immunomodularity, veterinary, human

## Abstract

Macrolide antimicrobial agents have been in clinical use for more than 60 years in both human and veterinary medicine. The discovery of the non-antimicrobial properties of macrolides and the effect of immunomodulation of the inflammatory response has benefited patients with chronic airway diseases and impacted morbidity and mortality. This review examines the evidence of antimicrobial and non-antimicrobial properties of macrolides in human and veterinary medicine with a focus toward veterinary macrolides but including important and relevant evidence from the human literature. The complete story for these complex and important molecules is continuing to be written.

## 1. Introduction

Macrolide antimicrobial agents have a long and successful history for the treatment of infectious diseases in humans and animals [1]. Erythromycin, the first discovered macrolide was isolated from the soil bacterium *Streptomyces erythraeus* and was first used clinically in 1952 [2]. Early macrolide use was considered as a suitable alternative to penicillin for penicillin allergic patients or in those with a penicillin resistant bacterial infection. Advanced generation macrolides were the result of chemical modifications to erythromycin and extended the spectrum of activity, have more favorable pharmacokinetic/pharmacodynamic characteristics and have fewer side effects [3].

Clinical uses of macrolides are extensive and varies between human and veterinary medicine. In humans, macrolides are indicated for the treatment of respiratory tract (i.e., community acquired pneumonia, *Bordetella pertussis* infection), skin and soft tissue, sexually transmitted and atypical mycobacterial infections. *Helicobacter pylori* gastrointestinal infections can be treated with clarithromycin in combination with amoxicillin, methronidazole and a proton pump inhibitor [4].

In veterinary medicine [5] macrolides are used to treat respiratory tract infections (i.e., bovine respiratory disease, swine respiratory disease), liver abscesses in cattle, foot rot in sheep caused by *Dichelobacter nodosus* [6] and for various infections in dogs and cats (dermatological, urogenital, respiratory tract and otitis media) and exotic animals [7]. Macrolides in food animals (cattle, swine) are used to treat infections but also for medication of large groups of animals—mass medication or metaphylaxis [8]. Table 1 summarizes select veterinary macrolides, clinical indicators and organisms on the drug label.

## 2. Classification

Macrolide classification is based on the lactone ring and the number of atoms and range from 12–16 members [16] with 12 membered ring compounds no longer in clinical use. Sugar moieties can be attached. Figure 1 compares the molecular structures of select veterinary macrolides. The first veterinary macrolide was spiramycin (1960s) [17,18] and for human use was erythromycin (1950s) (Figure 2). Semisynthetic macrolides appeared in human medicine in the late 1980s–1990s with the release of azithromycin [19]—an azalide—which is characterized based on a nitrogen atom(s) inserted into the lactone ring [19,20]. Clarithromycin was subsequently released and both azithromycin and clarithromycin have improved pharmacokinetic and pharmacodynamics properties compared to erythromycin [21,22,23,24,25]. Clarithromycin and the metabolite 14-hydroxyclarithromycin are active against a range of respiratory pathogens [26]. Tilmicosin is a 16-membered semi-synthetic derivative of tylosin and was developed exclusively for veterinary use. Gamithromycin was the 1st azalide approved for use in veterinary medicine [27]. Ketolides were the next advancement of the macrolide group and structurally are 14-membered macrolides with the L-cladinose moiety in position 3 replaced with a keto function [20]. Two compounds of note were telithromycin and cethromycin and both had activity against some macrolide resistant organisms—particularly *Streptococcus* spp. [28]. Tulathromycin is a semisynthetic molecule and is a 13 and 15 member ring macrolide with 3 amine groups and is termed a triamilide [29,30]. Tildipirosin is a 16-membered-ring semi-synthetic macrolide for veterinary use only [31].

## 3. Mechanism of Action

Macrolides, azalides, triamalides, and ketolides all share a similar mechanism of action. They block protein synthesis in susceptible bacteria by binding to the 23S rRNA on the large ribosomal subunit—50s—at the nascent peptide exit tunnel next to the peptidyl transferase center [3]. In essence, translation of mRNA is blocked as this prevents peptidyltransferase from adding amino acids from tRNA and elongation of growing peptide chains on the ribosome is halted and incomplete [32,33,34]. As binding is transient, the inhibition is reversible and macrolides are considered bacteriostatic and as such largely time dependent, however, at higher concentrations may display bactericidal properties [35,36]. Preventing reassembly of the ribosomes is a secondary mechanism of action.

## 4. In Vitro Activity

Generally, macrolides have potent activity against Gram-positive bacteria, non-enteric Gram-negative bacteria including fastidious genus/species and atypical organisms such as *Mycoplasma* species [37,38]. A summary of MIC data for 4 veterinary macrolides and key veterinary pathogens is shown in Table 2. While macrolides are not typically thought of being active against anaerobic bacteria Brook et al. indicated (from published studies) that macrolides have moderate to good in vitro activity against anaerobic bacteria other than *Bacteroides fragilis* group and *Fusobacterium* strains [39,40].

Macrolides have activity against some *Clostridium* species (e.g., *Clostridium perfringins*), lesser activity against *Fusobacterium* species and *Peptostreptococcus* species and activity against *Prevotella* species, *Porphyromonas* species, *Microaerophilic Streptococcus* species and non-spore forming anaerobic bacilli [52]. Macrolides do not have consistent activity against anaerobic Gram-negative bacilli. Clarithromycin specifically is active against oral cavity Gram-positive anaerobes (*Actinomyces* spp., *Propionibacterium* spp., *Lactobacillus* spp., *Bifidobacterium* spp.) whereas azithromycin is less active than erythromycin against these same organisms [52]. Azithromycin is the most active macrolide against anaerobic Gram-negative bacilli (e.g., *Fusobacterium* spp., *Bacteroides* spp., *Wolinella* spp., *Actinobacillus actinomycetemcomitans*).

For gamithromycin (www.zactran.com, accessed on 4 December 2022), minimum inhibitory concentration (MIC) values (µg/mL) for 50% (MIC_50_) and 90% (MIC_90_) and MIC range values of the strains tested were as follows: *Mannheimia haemolytica* (*n* = 85) 1, 1 and 0.5–>32; *Pasteurella multocida* (*n* = 79) 0.5, 1 and 0.12–>32; *Histophilus somni* (*n* = 32) 0.5, 0.5 and 0.25–1. For tulathromycin (www.draxxin.com, accessed on 4 December 2022) MIC_50_, MIC_90_ and range values (µg/mL) were as follows: *M. haemolytica* (*n* = 542) 2, 2, 0.5–64; *P. multocida* (*n* = 221) 0.5, 1, 0.25–64; *H. somni* (*n* = 36) 4, 4, 1–4; *Mycoplasma bovis* (*n* = 43) 0.125, 1, ≤0.063–>64, *Fusobacterium necrophorum* (*n* = 116) 2, 64, ≤0.25–>128; *Porphyromonas levii* (*n* = 103) 8, 128, ≤0.25–>128; *Actinobacillus pleuropneumoniae* (*n* = 135) 16, 32, 16–32; *Haemophilus parasuis* (*n* = 31) 1, 2, 0.25–>64; *Bordetella bronchiseptica* (*n* = 42) 4, 8, 2–8. For tildipirosin (www.zuprevo.com, accessed on 4 December 2022) MIC_50_, MIC_90_ and MIC range (µg/mL) were as follows: *M. haemolytica* (*n* = 484) 1, 2, 0.25–>32; *P. multocida* (*n* = 233) 0.5, 1, 0.12–>32; *H. somni* (*n* = 33) 2, 4, 1–4. For tilmicosin (www.micotil.com, accessed on 4 December 2022) MIC values were reported to be *M. haemolytica* 3.12 µg/mL, *P. multocida* 6.25 µg/mL, *H. somnus* 6.25 µg/mL, *Mycoplasma dispar* 0.097 µg/mL, *Mycoplasma bovirhinis* 0.024 µg/mL and *Moraxella bovoculi* 0.048 µg/mL. For testing of tilmicosin against organisms of human intestinal flora, the following MIC_50/90_ (µg/mL) values were reported; *B. fragilis* 16/128, *Bacteroides* spp. 8/64, *Bifidobacterium* spp. 0.125/8, *Clostridium* spp. 16/>128, *Enterococcus* spp. 8/16, *E. coli* 128/>128, *Eubacterium* spp. 32/32, *Fusobacterium* spp. 32/32, *Lactobacillus* spp. 16/>128 and *Peptostreptococcus* spp 32/32.

Blondeau et al. reported for 285 susceptible strains of *M. haemolytica* with MIC_50_, MIC_90_ and MIC range (µg/mL) values of 2, 8 and 0.25–8 for tilmicosin and 1, 2 and 0.125–2 for tulathromycin [53]. In a separate publication Blondeau and Fitch [41] reported the following MIC_50_, MIC_90_ and MIC range values (µg/mL) on susceptible strains: *A. pleuropneumoniae* (*n* = 67) 2, 4, 1–8 for tilmicosin and 1, 2, 0.5–8 for tulathromycin; *P. multocida* (*n* = 73) 2, 4, 1–8 for tilmicosin and 0.25, 0.5, 0.063–1 for tulathromycin; *S. suis* (*n* = 59) ≥4, ≥4, ≥4 for both tilmicosin and tulathromycin.

From a study investigating treatment of BRD with tulathromycin and ketoprofen, De Koster et al. [54] reported for 131 *P. multocida* isolates (93.38% susceptible) and 63 *M. haemolytica* strains (100% susceptible) an MIC_90_ of 2 µg/mL. The MIC values for *H. somni* ranged from 2–16 µg/mL and for *M. bovis* > 128 µg/mL.

## 5. Mechanism of Resistance

Macrolide resistance may be low level or high level [55,56]. Figure 3 is a schematic diagram summarizing antimicrobial resistance. Drug efflux or the active pumping of drug out of the bacterial cell by an efflux pump is due to a membrane bound efflux protein encoded by the *mef*(A) gene [57,58] and confers low level resistance. High level resistance follows ribosomal target site modification which is mediated by methylases encoded by the *erm*(B) gene which methylates adenine at position 2058 of domain V. This prevents macrolides from binding to the ribosome [59]. A third mechanism is conformational changes at the binding site resulting from mutations in the 23S rRNA. Macrolides may be enzymatically inactivated following hydrolysis of aglycone by esterases either through glycosylation or phosphyorylation of the 2′ hydroxyl group of their amino sugar [60,61].

## 6. Resistance Epidemiology

Macrolide susceptibility/resistance was determined for a number of bacterial pathogens associated with BRD in Alberta, Canada [48] (Table 2). Specimens were collected from both morbid and dead cattle. Of 233 *M. haemolytica* isolates (224 from lung samples), 44.2% were resistant to tilmicosin and 37.8% were resistant to tulathromycin. Of 226 *M. bovis* isolates (194 from lung samples) 98.2% were resistant to tilmicosin and 92% resistant to tulathromycin. Of 117 isolates of *P. multocida* (85 from lung specimens), 41.9% were resistant to tilmicosin and 29.9% were resistant to tulathromycin. For 75 isolates of *H. somni* (63 from lung specimens), 18.7% were resistant to tilmicosin and 21.3% were resistant to tulathromycin. Finally, for 94 isolates of *Trueperella pyogenes* (69 from lung samples), 75.6% were resistant to tilmicosin and 57.4% were resistant to tulathromycin.

In a 10 year study (2000–2009) of antimicrobial susceptibility of BRD pathogens from the USA and Canada testing *M. haemolytica* (*n* = 2977 over study duration and varying by year of collection and geographical location), *P. multocida* (*n* = 3291) and *H. somni* (*n* = 1844) Portis et al. summarized susceptibility/resistance trends for tilmicosin and tulathromycin plus several other drugs [49]. Considering *M. haemolytica*, susceptibility ranged from 59.5% to 89.4% for tilmicosin and 87.8% to 97.6% for tulathromycin with resistance rates to both agents being highest with isolates collected in 2009. For *P. multocida* MIC_90_ values ranged from 16–>64 µg/mL for tilmicosin compared to 4–16 for tulathromycin (90.9–96% susceptibility) with the higher MIC_90_ values being the later year of the study. With *H. somni*, MIC_90_ values for tilmicosin ranged from 16–64 and for tulathromycin (81–96% susceptibility) from 8–>64 with MIC_90_ values being higher in the 2008/2009 collected isolates. The authors concluded that MIC_50_ and MIC_90_ values declined (MICs increased) for tilmicosin and tulathromycin over the years the isolates were collected.

Macrolide resistance in Australia has been reported. Alhamami and colleagues commented on macrolide and tetracycline resistance with *M. haemolytica* and *P. multocida* from beef feedlots [47]. Isolates were collected from deep lung swabs post-mortem between 2014–2019 and tested against 19 antimicrobial agents. For *M. haemolytica*, only 1/88 (1.1%) was macrolide resistant. By comparison for *P. multocida* strains, 19/140 (13.6%) were resistant to tilmicosin and 17/140 (12.1%) resistant to tulathromycin/gamithromycin. Of 55 *M. haemolytica* isolates collected between 2014–2018, MIC_90_ values (µg/mL) were 1 for gamithromycin as compared to 8 for tilmicosin, 16 for tulathromycin and 1 for tildipirosin: for 33 isolates collected in 2019, the MIC_90_ (µg/mL) values for those same four drugs were 2, 8, 8 and 1, respectively. Of 75 *P. multocida* isolates collected between 2014–2018, MIC_90_ values (µg/mL) were 8 for gamithromycin as compared to 8 for tilmicosin, 8 for tulathromycin and 8 for tildipirosin: for 65 isolates collected in 2019, the MIC_90_ values for the 4 drugs were 16, 32, 64, 2, respectively. More so for *P. multocida*, there was a shift in MIC values (µg/mL) giving rise to higher MIC_90_ values (except tildipirosin) for the more recently collected isolates. While percent susceptible/resistant is a useful calculation, it may not sufficiently identify trends in the data and MIC_50_, MIC_90_ and MIC range values may be helpful. As shown in Table 2 and from the Australian study, MIC_50_ and MIC_90_ along with MIC range data showed a right shift toward less susceptible strains (higher MIC values but still within the susceptible designation) for some of the antibiotics summarized above. Whole genome sequencing identified *msr*(E) (enzyme inactivating 14-membered macrolides) and *mph*(E) (ATP transporter that transports erythromycin and streptogramin B from the cell using energy from ATP hydrolysis) encoding macrolide resistance in most isolates. As some strains were identified as multidrug resistant (i.e., macrolide, tetracycline and beta lactam) and the first such report in Australia for these pathogens, the authors stipulate close monitoring should now be done. In a report from 2011, McClary and colleagues reported from the USA that resistance to tilmicosin was uncommon being 0.8% (6/745) amongst strains of *M. haemolytica* and *P. multocida* (6.9%, 16/231) [62]. Additionally McClary et al. reported the proportion of treatment failures attributed to tilmicosin resistant (MIC ≥ 32 µg/mL) or non-susceptible (MIC ≥ 16 µg/mL), *M. haemolytica* was 0.2% and 0.5%, respectively.

Pyorala and colleagues commented on macrolide and lincosamide use in cattle and pigs and the development of antimicrobial resistance from a European perspective [5]. They indicated that resistance to macrolides/lincosamides have now emerged but understanding the true prevalence may be challenging due to lack of standards for susceptibility testing.

## 7. Pharmacokinetics

Macrolides typically have a large volume of distribution, are weak bases with high lipid solubility, good penetration to tissues and their activity is dependent on pH—higher than pH 7 gives optimal activity [20]. Macrolides accumulate in phagocytic cells and have high intracellular concentrations—higher than epithelial lining fluid and plasma drug concentrations [63]. In low pH environments (i.e., stomach) erythromycin is degraded whereas azithromycin and clarithromycin are more stable [64]. The relationship between high intracellular dug concentrations and bacterial killing still requires clarification [63,65]. Intracellular penetration and accumulation of antimicrobial agents in neutrophils has been argued to be important for killing or inhibition of intracellular pathogens. Additionally drug laden neutrophils attracted to the site of infection are thought to release drug thereby increasing drug concentration locally. Bongers et al. reviewed the intracellular penetration and effects of antibiotics on *Staphylococcus aureus* inside human neutrophils [66]. Specifically referring to macrolides (azithromycin, clarithromycin, erythromycin) and *S. aureus*, MIC range values were between 0.064–1 µg/mL and with neutrophil penetration (considering cellular/extracellular [C/E] ratio) being 4.4–34 and >100, the intracellular drug concentration to organism MIC is very favourable giving a high effect on the organism—even with a bacteriostatic effect. Azithromycin (>100) has a higher C/E ratio than does clarithromycin (9.2), erythromycin (C/E between 14–34) and other macrolides and lincosamides. Ahmad et al. reported that intracellular concentrations of azithromycin is high enough—especially in macrophages—to be effective against multiple strains/species of *Francisella* and that azithromycin treatment prolonged survival in an in vivo *Galleria mellonella* (larval stage), wax moth caterpillar model of infection [67]. Macrolides are eliminated primarily by the liver with some via bile as unchanged parent drug or drug metabolite [5]. Compound half-lives are variable and related to low clearance rates. In swine and cattle, the half-life or tulathromycin is ~96 h but shorter for gamithromycin (~48 h), Tildipirosin has a half-life > 96 h in swine and ~210 h in cattle.

Modric and colleagues reported on the pharmacokinetics and pharmacodynamics of tilmicosin in sheep and cattle [68]. In cattle the mean C_max_ value was 0.873 µg/mL as compared to 0.822 µg/mL in sheep and the time to maximum drug concentration was 0.5 and 3.9 h, respectively (*p* < 0.001). Regarding serum versus lung concentrations of tilmicosin, a study in rats infected with *Mycoplasma pulmonis* (compared to uninfected rats) showed a significant difference (*p* < 0.01) between serum and lung tilmicosin concentrations with the lung to serum ratio being 86:1 in non-infected rats and 178:1 in infected animals [69]. The half-life for tilmicosin was 29.4 h for cattle and 34.6 h in sheep [68]. According to Giguere et al. and Menge et al. measurable amounts of gamithromycin and tildipirosin can be detected for more than 2 weeks in plasma and 3–4 weeks in lung [15,70]. Foster et al. reported plasma, interstitial fluid and pulmonary epithelial lining fluid concentration for tulathromycin to be 1.82 µg/mL, 0.042 µg/mL and 0.87 µg/mL, respectively, following a 2.5 mg/kg dose in calves. Villarino et al. reviewed the pharmacokinetics of tulathromycin from a pulmonary perspective [71]. In plasma, the half-life ranges from ~2–~5 days being shorter in mice (<20 h) and longer in foals (~140 h).

In pigs the plasma C_max_ is 0.637 µg/mL with a terminal half-life of 75.6 ± 13.7 h. By comparison in beef calves, the C_max_ is 0.5 µg/mL with a terminal half-life of 90 h. From horses (50–71 days old), plasma tulathromycin drug concentrations ranged from 0.464–0.675 µg/mL and with a mean terminal half-life ranging from 105–140 h. For investigations measuring pulmonary drug concentrations, the middle and caudal lobes, PELF and BELF had tulathromycin concentrations ranging from 2.59 µg/mL to 7.89 µg/mL in pigs with terminal half-lives ranging from 87–213 h. In Holstein male calves, PELF and PELF cell drug concentration ranged from 3.73–19.5 µg/mL with a terminal half-life 270–330 h. Such data argues higher drug concentrations in pulmonary compartments than in plasma and with longer half-lives.

Parenteral administration of macrolides is associated with pain and inflammation at the injection site that can last for several days [72]. For example, in a study by Prohl and colleagues, subcutaneous infections of erythromycin in calves yielded palm-sized, painful swellings lasting several days and at necropsy, the injection sites showed extensive coagulation of connective tissue and musculature with granulation and edema. Interestingly, a similar reaction was not seen with azithromycin nor with non-macrolide antimicrobials investigated. Menge reported on the pharmacokinetics of tildipirosin in cattle [15]. Following a subcutaneous injection of 4 mg/kg, the maximum plasma concentration was 0.7 µg/mL.

## 8. Pharmacodynamics

Macrolides prevent bacterial growth by inhibiting protein synthesis [64] and may be bacteriostatic or bactericidal and exhibit time dependent killing. Liver injury is rare with macrolide treatment [73] and more common in human patients taking azithromycin versus erythromycin. Comparatively, clarithromycin has better in vitro activity against Gram-positive organisms while azithromycin has better activity against Gram-negative bacteria. Azithromycin and clarithromycin have better activity against Gram-positive and Gram-negative bacteria than does erythromycin and fewer side effects [74]. Macrolides in general (including those used in veterinary medicine) typically have better in vitro (by MIC measurements) activity against Gram-positive and atypical organisms (i.e., *Mycoplasma* spp.) and adequate activity against select Gram-negative organisms [21].

## 9. Inflammation

Inflammation is a natural response to infection or injury. Simplistically, inflammation localizes to the site of infection/injury with a coordinated cellular response to eliminate, i.e., “the pathogen” along with damaged tissue/cells as a component of repair and healing. Short term acute inflammation is seen as essential for animal survival in response to injury or infection [75]. Chronic inflammation may increase with age and can be associated with metabolic disturbances, type 2 diabetes, cardiovascular disease, chronic kidney disease, cancers, autoimmune diseases, neurodegeneration diseases, osteoporosis, arthritis and other chronic conditions [75]. According to Furman and colleagues, “…certain social, environmental and lifestyle factors can promote systemic chronic inflammation…” leading to several diseases.

A cytokine storm can be defined as a severe immune response where an over abundance of cytokines are released in the blood stream too quickly. While some disagreement exists over the exact definition of a cytokine storm, Fajgenbaum and June indicate that under this all-encompassing term, systemic inflammation and multi-organ dysfunction failure occur, which in turn, can lead to death [76].

In food animals where various growth or production parameters are measured, along with animal health, Buret indicated growth, food intake, reproduction, milk production and periparturient metabolic health are all negatively affected by inflammation [77]. Immunological markers associated with physiological parameters in food animals include tumour necrosis factor-alpha, interleukins, insulin-like growth factor-1, cortisol, prostaglandin F2-alpha and changes in plasma calcium levels.

Collectively the inflammatory response is a rapid and robust cascade which if not self-limited could be lethal or if it becomes chronic can cause a range of life long chronic conditions.

## 10. Anti-Inflammatory Properties

### 10.1. Studies in Humans

Immunomodulatory effects of macrolides were described shortly after their introduction to clinical use in the 1950’s [78,79,80,81]. Indeed, Plewig and Schopf studied in vitro the anti-inflammatory effects of various antimicrobial agents including erythromycin in experimentally induced follicular pustules and found that topical erythromycin (and other drugs) lead to suppression of experimentally induced inflammation [82]. Many accept that the landmark study of Kudoh and colleagues on improved survival of patients with diffuse panbronchiolitis treated with low-dose erythromycin as solidifying the importance of the non-antibacterial properties of macrolides for patient care [83]. Table 3 is a summary of various immunological markers, their function and the effect of macrolides.

Kudoh and colleagues reported on the clinical effect of low-dose, long-term erythromycin chemotherapy on diffuse panbronchiolitis [132]. Diffuse panbronchiolitis is a progressive inflammatory condition of the airways and found exclusively in Japan [133]. The 5 year survival rate improved from 62.9–91.4% [134]. Indeed, macrolide therapy has non-bacterial impacts in patients with chronic airway diseases including chronic bronchitis, cystic fibrosis, bronchial asthma, chronic rhinosinusitis and bronchiectasis. Spagnolo and colleagues [135] indicated the macrolides regulate leukocyte function, production of inflammatory mediators and resolution of inflammatory, control of mucus hypersecretion and modulation of host defenses. Suppression of virulence factors are also a characteristic of macrolides [136].

Cystic Fibrosis (CF) is an inherited disease primarily affecting the lungs and digestive system. It is characterized by sticky, thick mucus, persistent lung infections leading to lung destruction and loss of function, eventually leading to death [137]. Sinus infections may be coincidental with CF in some patients. Macrolides have been investigated in CF patients. Saiman et al. investigated azithromycin in 185 CF patients > 6 year of age and that had chromic *Pseudomonas aeruginosa* infections [138]. Patients were randomized to have azithromycin at either 250 or 500 mg (based on body weight) or placebo 3 times/week for 24 weeks. *Pseudomonas aeruginosa* strains are intrinsically resistant to azithromycin based on in vitro MIC measurements and susceptibility breakpoint. Imperi et al. [136] stated “…azithromycin provides a paradigmatic example of an “unconventional” antibacterial drug. Besides it’s growth-inhibiting activity, AZM displays potent anti-inflammatory properties, as well as antivirulence activity…” At enrollment, patients had a forced expiratory volume in one second (FEV_1_) of >30% of predicted value. Patients in the azithromycin arm had a 4.4% improvement on their FEV_1_ as compared to a 1.8% decrease in patients in the placebo arm—a highly significant difference (*p* = 0.001). Additionally, azithromycin treated patients saw improvement in their forced vital capacity (FVC), they gained more weight (*p* = 0.02) and were less likely to have an exacerbation (*p* = 0.03), had fewer hospital days and intravenous antibiotics than were patients in the placebo arm. Other studies in CF patients have also shown a beneficial effect of azithromycin therapy. For example, Pirazda and colleagues in a retrospective study in adults given 250 mg OD and followed for 21 months showed an improvement in FEV_1_, FVC, weight gain and a decrease in intravenous antibiotics [139]. Equi and colleagues from a randomized double-blind placebo-controlled crossover trial (250–500 mg OD) involving 41 children in 2 × 6 month blocks reported improvement in FEV_1_ and a decrease in oral antibiotics [112].

Howe and Spencer commented on macrolides for treatment of *P. aeruginosa* infections [140]. Based on in vitro susceptibility testing, minimum inhibitory concentration (MIC) values typically range from 128–512 µg/mL (or higher). In considering erythromycin and a 250 mg oral dose, peak serum concentrations range from 1–1.5 µg/mL and sputum concentrations following a 1 g IV dose every 12 h was 2.6 µg/mL. Clarithromycin at a 250 mg dose was reported to have a C_max_ of 1.1 µg/mL, an epithelial lining fluid concentration of 26.1 mg/mL and an alveolar macrophage concentration of 222 mg/mL [141]. As such, is this a paradoxical observation given that achievable and sustaining macrolide drug concentrations appear insufficient to have an antimicrobial effect with *P. aeruginosa*. Howe and Spencer argue two main mechanisms may be responsible for these observations: (1) immunomodulatory effects on the immune response and (2) a direct effect on *P. aeruginosa* to decrease virulence [140]. They summarize macrolides concentrate in immune effector cells [142], stimulate chemotaxis, increase cytokine production and bactericidal activity and inhibit superoxide anion production by neutrophils [143,144,145,146]. Regarding impacting virulence, macrolides at clinically relevant concentrations of 0.5–10 µg/mL have been reported to inhibit exotoxin A, protease, elastase, phospholipase C, DNase, lecithinase, gelatinase, lipase, pyocyanin and motility [147,148,149,150,151,152,153]. Finally, macrolides might not all be the same and Howe and Spencer indicate that azithromycin is more inhibitory than is clarithromycin which is more inhibitory than is erythromycin and that variability may exist between different bacteria and susceptibility of different virulence factors to macrolides. Regardless, the data appears clear regarding the non-antimicrobial effects of macrolides.

Bronchiectasis is a chronic pulmonary condition characterized by thickened walls and dilatation of the bronchi resulting from infection and inflammation [154]. Two studies with azithromycin [155,156] (500 mg twice weekly or 250 mg 3 times weekly) in adults and evaluated over 6 and 10 months showed a decreases in exacerbation frequency, a decrease in sputum volume (500 mg twice weekly) a decrease in IV antibiotics and symptoms (250 mg 3 times weekly). Koh et al. reported a decrease in airway responsiveness and the FEV_1_ being unchanged in 25 children treated with 4 mg/kg twice daily of roxithromycin and evaluated for 12 weeks [157]. Tsang et al. reported on 21 adults treated with erythromycin (500 mg twice daily) for 8 weeks and the benefit included improved FEV_1_, and FVC and decreased sputum volume [158]. For 34 children treated with clarithromycin 15 mg/kg and followed for 3 months, decreased sputum volume, improved FEF_25–75_ (maximal mid expiratory flow) and an unchanged FEV_1_ was reported by Yalcin and colleagues [117].

Chronic obstructive pulmonary disease (COPD) refers to a group of diseases that result in blockage of airflow leading to breathing difficulties in affected individuals. Emphysema and chronic bronchitis are included in COPD related diagnosis. Emphysema affects alveoli thereby interfering with the transfer of oxygen and carbon dioxide due to the alveoli being abnormally inflated. Chronic bronchitis is a long-term condition in which bronchi become inflamed and ultimately scarred resulting in large amount of mucus production, chronic cough and difficultly with breathing. At least 2 studies have reported on the impact of macrolide therapy and COPD. Banerjee and colleagues in a randomized prospective double blind placebo controlled trial investigated 500 mg once daily of clarithromycin for 3 months in 67 adults [159]. Overall, the patient’s health status was unchanged, the exacerbation rate was unchanged and the sputum bacterial numbers remained unchanged but improvements in symptom scores and physical function scores were seen. Gastrointestinal upset was reported. In a randomized double-blind placebo-controlled trial with erythromycin (250 mg twice daily) in 109 adults over 1 year, Wilkinson et al. reported a decrease in the incidence of COPD related exacerbations [160]. Patients in the placebo arm were significantly (*p* = 0.004) more likely to be treated for an exacerbation than those in the erythromycin group.

Chronic rhinosinusitis can be broadly defined as an inflammatory disease of the paranasal sinuses and is usually defined by the presence of 2/4 symptoms—facial pain/pressure, hyposmia/anosmia, nasal drainage and nasal obstruction—for at least 12 consecutive weeks [161]. Cervin et al. reported on one-year, low dose daily erythromycin/clarithromycin treatment for persistent chronic sinusitis following surgery [162]. Seventeen patients were investigated of which 12 responded to treatment. At 12 month follow up, improvements in saccharine transit time (*p* < 0.05) but not ciliary beat function were seen. A trend toward an increase in nasal nitric oxide was seen but not statistically significant (*p* = 0.12) and endoscopic nasal examination scoring improved (*p* < 0.01). Using a visual analog scoring scale, pronounced improvements (*p* < 0.01) were seen in nasal congestion, sticky secretion and runny nose and also improvements in headache (*p* < 0.05). The authors concluded that long term, low dose erythromycin treatment is effective in persistent chronic sinusitis that does not respond to sinus surgery or systemic steroid or antibiotic treatment, however, placebo controlled trails were needed. Another study by Wallwork et al. investigated macrolide therapy in chronic rhinosinusitis in a double-blind, randomized, placebo controlled trial involving 64 patients administered either 150 mg roxithromycin daily for 3 months or placebo [97]. Significant improvements were seen with the Sinonasal Outcome Test-20 (SNOT-20) score [163], nasal endoscopy, saccharine transit time and interleukin-8 levels in lavage fluid (*p* < 0.05) in the macrolide treatment group. A correlation was detected between improved outcome measurements and IgE levels. Improvements in olfactory function, peak nasal inspiratory flow or fucose and alpha 2-macroglobulin from lavage fluid was not seen. Placebo arm patients did not show any improvement in any outcome measurement. The authors concluded macrolides may have a beneficial role for treatment of chronic rhinosinusitis and additional studies are required to define the role of macrolides in clinical practice.

Asthma is a long term condition of children and adults in which airways narrow and swell and may be associated with extra mucus production. This leads to symptoms such as coughing, wheezing and shortness of breath. Asthma attacks may be triggered by medicines, weather changes (thunderstorms, high humidity, cold dry air), foods and food additives, fragrances, tobacco smoke, dust mites, air pollution, pests, pets, mold and cleaning and disinfection (cdc.gov/asthma/triggers.html, accessed on 4 December 2022). Macrolide agents have been investigated for potential immunomodulatory/anti-inflammatory effects in asthma. Three randomized, double-blind, placebo controlled trails with clarithromycin (500 mg bid [105], 250 mg bid/tid [164], 500 mg bid [116]) enrolled between 45–63 adult patients with study durations ranging from 6–8 weeks and inhaled corticosteroids may or may not have been used. In 2/3 studies FEV_1_ remained unchanged (one improvement was seen in patients PCR positive for *Mycoplasma pneumoniae* or *Chlamydia pneumoniae*), improved quality of life and decreased wheezing was reported in one study. Bronchial hyper-responsiveness (increase in sensitivity to airway narrowing stimuli) was unchanged in one study and decreased in another. In a randomized, double-blind, placebo controlled crossover study with clarithromycin 200 mg bid for 8 weeks in 17 adult patients without inhaled corticosteroids, Amayasu and colleagues reported a decrease in symptom score and bronchial hyper-responsiveness but no change in FEV_1_ [104]. Johnston et al. investigated telithromycin 800 mg daily for 10 days in 278 adult patients for 8 weeks in a randomized, parallel group, double blind placebo controlled trial [165]. A decrease in symptom score was observed in telithromycin treated patients (*p* = 0.004) but not in placebo patients. Home peak expiratory flow was unchanged. Some 61% of patients had evidence of infection with either *C. pneumoniae*, *M. pneumoniae* or both, but there did not appear to be any relationship between bacteriological status and asthma treatment responses. The mechanisms of the benefits of telithromycin in asthma patients remained unclear. Two reports [107,166] investigated roxithromycin 150 mg bid: 1 with 232 adults for 6 weeks with >75% using inhaled corticosteroids and the other with 14 adult patients without inhaled corticosteroids in 2 × 8 week blocks in a crossover study. Both studies were randomized, double blind and placebo controlled. In the first study, the morning peak inspirator flow was unchanged but increased in the evening; symptom score was unchanged [166]. In the 2nd study, FEV_1_ and bronchial hyper-responsiveness was unchanged but there was a decrease in symptom score [107]. In fact, blood eosinophils, serum eosinophilic protein, sputum eosinophils and sputum eosinophilic protein were significantly decreased in roxithromycin treated patients. Kamada et al. evaluated the efficacy of low-dose troleandomycin with and without steroids in 18 children with severe steroid requiring asthma [167]. There were three treatment arms (randomized, double blind, parallel fashion) which included troleandomycin and methylprednisolone, troleandomycin and prednisone and methylprednisolone alone. All 3 groups tolerated significant (*p* < 0.05) reductions in glucocorticoid dose over the 12 weeks of the study being highest (80% ± 6%) for troleandomycin and methylprednisolone and lowest (44% ± 14%) for methylprednisolone alone. A statistically significant difference (*p* < 0.05) was also seen between the troleandomycin and methylprednisolone and methylprednisolone alone groups. A reduction in symptom score was also noted in the troleandomycin/methylprednisolone group. Pulmonary function remained unchanged regardless of group treatment. The authors concluded troleandomycin was safe and might be an alternative treatment for patients unable to tolerate tapering of their steroid dosing, although it is acknowledged the study sample size was small. In another study with troleandomycin [168], 75 patients with asthma requiring daily corticosteroids, were enrolled in a 2 year trial (double blind, placebo controlled) comparing troleandomycin with methylprednisolone versus methylprednisolone alone for asthma therapy with the primary outcome being the lowest stable steroid dose and steroid side effects. No benefit was acknowledged in this study.

Thomas and Gibson [169] in a recent editorial commented on a study by Ghimire and colleagues [170] which evaluated the efficiency of low dose (10 mg/kg, 3 × week) long-term (3 months) azithromycin therapy in children with uncontrolled asthma. This study from India enrolled 120 children (60 per group) with a mean age of 9.9 ± 3.0 years. Children in the azithromycin group had worse asthma control than did the control group (*p* = 0.015). A significant (*p* < 0.001) improvement in asthma control was seen in the azithromycin group following 12 weeks of therapy. Scoring based on the Childhood Asthma Control Test (CACT) or Asthma Control Test (ACT) showed a substantial improvement over the control group at 12 weeks of follow up (*p* < 0.001). Secondary outcomes such as the number of acute exacerbations precipitating an emergency room visit or steroid use were also improved. Spirometry parameters, fractional exhaled nitric oxide and percentage of neutrophils in sputum were not affected by azithromycin therapy. In their editorial, Thomas and Gibson indicated the findings by Ghimire et al. are encouraging but raise a few questions including optimal azithromycin treatment regimes are not established in adults or children, the lack of consistency between macrolides in previous studies, the absence of head to head macrolide studies, different macrolide dosages, ideal duration of treatment and if asthma control was sustained after discontinuation of therapy. Regardless, the authors acknowledge the positive effects observed encourages more research to address important questions such as the most effective macrolide, ideal therapeutic regimen, duration of treatment and long-term consequences (if any) of long-term use and the mechanism(s) of action.

Ci et al. in a non-infectious mouse model of asthma reported on the anti-inflammatory effects of tilmicosin [171]. Tilmicosin treatment showed a marked reduction in the presence of some immune cells and cytokines from BAL specimens. Specifically, tilmicosin at 10 and 20 mg/kg significantly reduced eosinophils, neutrophil and macrophage infiltration and was dose dependent (*p* < 0.05 and *p* < 0.01) when comparing untreated challenged mice. Regarding cytokines, tilmicosin (dose dependent) significantly (*p* < 0.05, *p* < 0.01) inhibited IL4, IL5 and IL13 production at 20 mg/kg and reduced inflammatory cell infiltration in peribronchial and perivascular areas. Other tilmicosin effects included reduced mucus secretion, goblet cell hyperplasia, epithelial cell disruption and reduced airway resistance. The authors summarized that efficacy of tilmicosin and other macrolides may be related to anti-inflammatory and immunomodulatory properties in addition to anti-bacterial activity.

Lung transplantation is associated with some potential long term health complications. Bronchiolitis obliterans syndrome (BOS) is a rare chronic pulmonary disease that deteriorates over time in lung transplant recipients [172]. Hakim and colleagues described BOS as “… devastating manifestation of chronic graft-versus-host-disease…” [173]. It is characterized by damaged bronchioles that become inflamed, leading to scarring and blocked airways. Morbidity and mortality are high. Recommended therapies include a macrolide—azithromycin. Some 5 prospective, open label [174,175,176,177,178] and 1 retrospective case series [179] study investigated azithromycin therapy for BOS. The number of patients in each study ranged from 6–14 (20 in the retrospective case study) and treatment times ranged from 12 weeks to 10 months. Dosages included 250 mg daily for 5 days and then 3 times weekly, 250 mg daily for 5 days then on alternate days, 250 mg on alternate days, 500 mg qid for 3 days followed by 250 mg 3 times weekly. Notable responses were improvement in FEV1 (significant in some patients), improved forced vital capacity and in one study some patients had higher neutrophil levels and higher interleukin 8 levels. One study reported no improvement in lung function.

### 10.2. Studies in Veterinary Medicine

Piccinno and colleagues commented on the “unconventional” effects of antimicrobial agents in bovine reproduction and looked at various drug classes [180]. Specifically referring to macrolides, there were both uterine contraction and anti-inflammatory effects. For example, Granovsky-Grisaru et al. studied the impact of erythromycin on contractibility of rat myometrium [181]. In this study, myometrial strips from pregnant rats were suspended in tissue baths and contractions monitored. Erythromycin caused a sustained decrease (amplitude reduced to 22% of control and frequency reduced to 38% of control) in phasic contractions induced by either oxytocin or carbachol. Celik and Ayar reported on inhibition of human myometrial contractions in isolated human myometrium by clarithromycin [182]. Myometrium strips acquired during cesarean section were stimulated with oxytocin, prostaglandin F_2alpha_, and potassium chloride induced contractions and the impact of clarithromycin which inhibited the amplitude of contraction and was dose dependent.

Abdel-Motal investigated the immunomodulatory effects of tulathromycin in rabbits [183]. Specifically, these investigators tested tulathromycin alone or in combination with vitamin C on immunomodulating apoptotic effect and DNA of some immune cells and to investigate the effect on serum antioxidants activity. According to Renehan and colleagues [184] “Apoptosis describes the orchestrated collapse of a cell characterized by membrane blebbing, cell shrinkage, condensation of chromatin and fragmentation of DNA followed by rapid engulfment of the corpse by neighboring cells”. Antioxidant substances prevent or slow damage to cells by free radicals. In this study, 25 healthy, 3 month old rabbits were divided into 5 groups of 5 animals each: group 1 was control, group 2 was vaccinated with 1 mL/rabbit subcutaneously of *P. multocida* vaccine, group 3 was injected with 17.5 mg/rabbit of vitamin C and 1 mL/rabbit of *P. multocida* vaccine, group 4 was given 1 mL/rabbit of *P. multocida* vaccine and 2.5 mg/kg body weight (BW) of tulathromycin and group 5 was given 17.5 mg/kg vitamin C and 1 mL *P. multocida* vaccine and 2.5 mg/kg BW of tulathromycin. Tulathromycin induced significant decrease in lysozyme levels (~9–10%) and phagocytic activity (~5.7–7.7%) on days 1–3 post vaccination and a significant decrease in lymphocyte transformation (49.1%) on day 3 but the addition of vitamin C with tulathromycin elicited an increase in lysozyme levels (~13.9–15.8%) and phagocytic activity (9.2–9.7%) on days 1–3 and an increase in lymphocyte transformation (158.6%) on day 3 post vaccination compared with the tulathromycin group. When the impact on humoral immune responses was measured, tulathromycin induced a non-significant reduction in total globulins and a significant decrease in antibody titer (5.4% on day 7 and ~7% on days 14 and 21 post vaccination) compared to the vaccinated group and the addition of vitamin C with tulathromycin yielded a non-significant increase in total globulins and a significant increase in antibody titer (5.7% on day 7 and 6.7% on day 14 and 9.4% on day 21 post vaccination) when compared to the tulathromycin group. The authors suggest the observed effects with tulathromycin were potentially related to macrolide inhibition of pro-inflammatory cytokines including interleukin (IL) 1, IL6, IL8 and tumor necrosis factor -Alpha via suppression of the transcription factor nuclear-ĸB or the activator protein-1 [185]. Macrolides are also known to suppress neutrophil function by interference of: first, IL-8 production and TNF-alpha by structural cells and macrophages; second, decreased expression of adhesion molecules on neutrophils and the vascular endothelium and third reduced production and release of enzymes by neutrophils [109].

Urban-Chmiel and colleagues investigated the effect of 3 macrolides—tylosin, tilmicosin and roxithromycin—on the properties of bovine leukocytes [84]. The aim of the study was to investigate viability, chemotaxis, apoptosis and oxidative stress in bovine leukocytes in vitro. The authors argue macrolides weaken the inflammatory response by inhibiting IL-1, IL-2 and TNF-alpha and also modify immune cell activity altering phagocytic process functions [186,187]. Blood harvested from 60 disease free 1 week–2 year old Holstein-Friesian cows were divided into 4 groups: 1, calves < 1 month, group 2 > 1 month and <5 months, group 3 calves between 6–9 months and group 4 > 12 months. All macrolide antibiotics reduced bovine leukocyte viability from calves up to one month (control 98% versus 96.5–97.5% in the presence of the drugs) of age. From calves between >1 month to 5 months (leukocyte viability between 89.8), leukocyte viability in the presence of the macrolides was reduced to 84.9% with the lowest viability in the presence of roxithromycin (84.9%); from animals between 6–9 months, viability (95.2%) was reduced to 92.3–93.8%; cattle over 1 year (95.9%) did not show substantial reductions in viability (93.9–95.7%) as compared to control. In comparing lymphocytes, monocytes and neutrophils, the effect of the macrolide compounds was lesser on lymphocytes with viability remaining between 91.9–99.2% across all age groups, however, a non-statistical decrease in viability was seen with tilmicosin exposed animals in the >1–5 month old calves. Monocytes viability in the >1–5 month age group decreased to 42.7% in the presence of tylosin. For neutrophils (75.5–78.6% for controls across all 4 groups) the change in viability in the presence of macrolides was variable being increased in the presence of roxithromycin in group 1 and reduced in the presence of tylosin in group 4. Viability in the presence of tilmicosin varied form 69.2–82.4% across all 4 groups. Cellular metabolic activity was determined using the nitro blue tetrazolium reduction (NBT) assay and measurements of absorbance. Tilmicosin and roxithromycin had the greatest non-significant decrease in absorbance with leukocytes from calves between >1–5 months. Nitrate ion (*NO*) concentrations were not significantly affected by any of the macrolides tested, however, *NO* concentrations in cells were affected by the antibiotics tested with the strongest correlation seen with leukocytes incubated with tilmicosin. Apoptosis was significantly (*p* ≤ 0.5) affected by the macrolide tested with the lowest percentage of apoptotic cells observed in cells from group 1 incubated with roxithromycin (2.1%). The highest number of apoptotic cells was seen in cells from calves >1–5 months of age incubated with tilmicosin and was significantly (*p* ≤ 0.05) different form control. Significant differences were also seen across all groups for leukocytes incubated with roxithromycin. Macrolides had an impact on leukocyte chemotaxis resulting in a reduction in chemotactic activity—regardless of age of animal in each group. Average chemotactic leukocyte activity in the presence of drugs ranged from 49–76.3% with the highest significant (*p* ≤ 0.05) decrease in migration was seen with cells from calves aged 1–>5 months and incubated with tylosin and the smallest difference in chemotaxis seen with leukocytes from cattle from 6–9 months and incubated with tylosin or tilmicosin. The authors concluded all investigated macrolides exhibited a modulatory effect on leukocyte functions from cattle of all ages.

Zimmerman and colleagues performed a systemic review of the immunomodulatory effects of macrolides and the underlying mechanisms [188]. They argued that the mechanism underlying the non-antibacterial effects were not well understood. Their literature review covered publications from 1946–2016 and focused on azithromycin, clarithromycin, erythromycin and roxithromycin. Some 47 different immunological markers were investigated from over 44 publications (22 randomized controlled trials, 16 prospective cohort and 8 case–control studies) from 17 countries. Then, clinical conditions investigated included blepharitis, periodontitis, nasal polyps, rhinosinusitis, asthma bronchiale, bronchiectasis, chronic obstructive pulmonary disease, diffuse panbronchiolitis, cystic fibrosis, lung transplantation, diabetic nephropathy, coronary atherosclerosis and healthy volunteers. Markers investigated included various interleukins (IL-1beta, IL 2, 4, 5, 6, 8, 12, 17), total cell, leukocyte neutrophil, macrophage, eosinophil and thrombocyte counts, neutrophil oxidative burst, chemotaxis, lactoferrin and elastase, eosinophilic cationic protein, tumor necrosis factor alpha, interferon gamma, transforming growth factor beta, granulocyte macrophage colony-stimulating factor, vascular endothelial growth factor, matrix metalloproteinase-9, E-selectin, C reactive protein and serum amyloid A. Summarizing from their comprehensive review, the authors indicated that overall, there was a more often decrease in immunological markers versus an increase or no changes in immunological markers in the presence of macrolides. The most frequently reported macrolide-induced changes were decreases in interleukin concentrations, neutrophil counts, TNF-alpha, neutrophil elastase, eosinophilic cationic protein, matrix metalloproteinase 9 and oxidative bust activity. Interestingly, of the macrolides compared, azithromycin was more frequently associated with no influence on immunological markers.

Otsuka et al. [189] reported on tulathromycin and the inhibition of endotoxin activity in endotoxemic calves. Endotoxin is a potent inducer of the inflammatory response and is a component of Gram-negative bacteria cell walls [190]. They argue the inhibition of cytokines with macrolides is known but not the effect on endotoxin. Ten healthy calves (30.9 days old and 40.9 kg weight) received either saline (*n* = 5) or 2.5 mg/kg tulathromycin 4 times every 3 days to maintain blood concentrations. Calves subsequently received 2.5 µg/mL of lipopolysaccharide via an indwelling jugular vein catheter on the day when tulathromycin blood concentrations stabilized. The peak endotoxin activity, in vivo, in the tulathromycin group was significantly (*p* < 0.05) lower than in the control group. The authors concluded that tulathromycin significantly lowered endotoxin activity and that prophylactic use may reduce symptom severity if cattle develop endotoxin-associated inflammation. Crosbie and Woodhead identified 14 and 15 membered macrolides as having immunomodulatory effects [133]. Er and Yazar used a rat model of Lipopolysaccharide (LPS)—induced lung injury to measure the impact of tylosin, tilmicosin and tulathromycin on inflammatory mediators (TNF, IL-1β, IL-6, IL-10, C-reactive protein (CRP), 13,14-dihydro-15-keto prostaglandins F_2_ alpha) in BAL fluid [191]. Area under the curve and maximum plasma concentration of the various markers was measured. Tylosin and tilmicosin decreased TNF, tilmicosin decreased IL-10, all 3 compounds decreased CRP, tylosin and tulathromycin decreased prostaglandin. Equine asthma due to neutrophilic inflammation is thought to contribute to airway obstruction and remodelling. It is characterized by bronchospasm mucus accumulation, luminal neutrophilic and airway remodelling [192]. Mainguy-Seers et al. investigated azithromycin in horses with severe asthma to determine any effects on improvement in lung function, mucus accumulation and central airway remodelling by decreasing luminal neutrophilia [87]. In this evaluation, azithromycin therapy was compared to that of ceftiofur using a 10 day regimen in 6 horses aged 13 ± 2.5 years and weighing 534 ± 53 kg. This study was a cross-over design whereby 3 horses received 10 mg/kg orally of azithromycin for 5 days then every other day for 2 additional doses; 3 horses received ceftiofur crystalline tree acid 6.6 mg/kg twice at 4 days apart. The washout period was for 3 weeks with diet and turnout remaining the same following which treatments were inverted. Evaluation of lung function, tracheal mucus accumulation, tracheal wash bacteriology, bronchial remodelling, airway neutrophilia and mRNA expression of proinflammatory cytokines was from bronchoalveolar lavage fluid. No effect was seen in horses treated with ceftiofur. Azithromycin therapy was associated with a decreased expression of IL-8 (*p* = 0.03, one-tailed) and IL-1β (*p* = 0.04, one-tailed) but no effect on the other variables measured. The authors indicated that antibiotic monotherapy for this condition is not justified.

Morck reported tilmicosin therapy for experimentally induced pulmonary pasteurellosis in young calves [193]. A study of 24 healthy Holstein calves (<70 kg) had intrabronchial inoculation (4 × 10^9^ CFUs) of *Pasteurella* (*Mannheimia*) *haemolytica* following which they received treatment with saline (*n* = 12) or with a single injection of 10 mg/kg BW of tilmicosin (*n* = 12). Tilmicosin treated calves were clinically better at 8 h post-treatment compared with saline treated (*p* < 0.05) and at necropsy, tilmicosin treated calves had significantly less severe gross and histological lesions (*p* < 0.05). Chris and colleagues investigated tilmicosin and the impact on apoptosis in bovine peripheral neutrophils in the presence of *P. haemolytica* [88]. The drugs investigated included tilmicosin, penicillin, ceftiofur, oxytetracycline, dexamethasone plus a phosphate-buffered saline (PBS) control. Neutrophils exposed to tilmicosin became apoptic (*p* < 0.01 versus PBS control)—regardless of the presence or absence of *P. multocida* but this was not seen with the other drugs. As well, tilmicosin treated apoptic neutrophils were phagocytosed at a significantly greater rate by bovine macrophages then were PBS treated controls (*p* < 0.01 versus controls at 30 min, 1 h and 2 h comparisons). The authors concluded tilmicosin (regardless of the presence or absence of *P. haemolytica*) induced neutrophil apoptosis and promotes phagocytic clearance of dying inflammatory cells thereby providing clinical benefits associated with severe inflammation and not specifically related to antibacterial effects.

Bovine mastitis may transition from acute to chronic and *Staphylococcus aureus* is a major pathogen in both clinical and sub-clinical presentations [194]. Casein synthesis occurs in the mammary gland with a biological function being primarily nutritional. Martinez-Cortes et al. investigated the potential for tilmicosin to modulate innate immune responses and preserve casein production in bovine mammary alveolar cells during infection with *S. aureus*. This was an in vitro study using immortalized mammary epithelial cells and an American Type Culture Collection (ATCC) strain of *S. aureus* (ATCC #27543). These authors reported tilmicosin decreased intracellular infection (*p* < 0.01), had a protection effect on the mammary epithelial cells reducing apoptosis after infection by 80% (*p* < 0.01), reduced reactive oxygen species (*p* < 0.01), IL-1β (*p* < 0.01), IL-6 (*p* < 0.01) and TNF-α (*p* < 0.05). Mitogen-activated protein kinase phosphorylation, measured by fluorescent-activated cell sorting was done to investigate the signaling pathway(s) in the immunomodulating effect of tilmicosin. Pre-treatment with tilmicosin increased extracellular signal-regulated protein kinase (ERK)½ (*p* < 0.05) but decreased P38 phosphorylation (*p* < 0.01). Casein synthesis in mammary epithelial cells was preserved (*p* < 0.01) in the presence of tilmicosin.

Lipopolysaccharide (LPS) is a potent inducer of the inflammatory response [195]. Reuter and colleagues reported on the effects of dietary energy source and tilmicosin on immune function in LPS challenged beef steers [196]. In this study, 24 Angus and Hereford cross-bred steers—weight 247 kg ± 2.4 kg—were used. Tilmicosin treatments were compared to physiological saline. The other variable in this study was diet. Blood samples were collected on day 0 and thereafter on days, 7, 14, 21, 27, 28, 29, 30, 31, 35, 43 and 49. On day 28, steers were challenged with 2.0 mg/kg BW of LPS from *E. coli* 0111:B4 in a bolus of 2.4–2.8 mL of physiological saline. Increased energy intake (70% concentrate diet ad libitum) increased (*p* ≤ 0.4) dry matter intake, average daily gain and rectal temperature after challenge. The 30% concentrate diet (fed ad libitum) increased pro-inflammatory cytokines—interferon-gamma, TNF-α and IL-6 (*p* ≤ 0.05)—compared to other diets. Decreased energy intake increased IL-6 (*p* ≤ 0.003) after LPS administration. Tilmicosin decreased the time to attain maximal rectal temperature (*p* = 0.01) and interacted with energy intake to increase pre-challenge pro-inflammatory cytokines (*p* ≤ 0.05). The effect of temperature, nutrition and health on immunity and immune function has been previously reported [197,198].

Moges et al. indicated excessive accumulation of neutrophils followed by uncontrolled death by necrosis at the inflammation site exacerbates inflammatory responses leading to tissue injury and decrease or loss of organ function [199]. Tylvalosin (macrolide derived from tylosin) was investigated for anti-inflammatory and pro-resolution benefits. This study involved neutrophil and monocyte-derived macrophages collected from fresh blood samples for 12–22 week old pigs. Cell exposures were to vehicle or tylvalosin following which apoptosis, necrosis, efferocytosis (removal of apoptic cells by phagocytic cells) and production of cytokines and lipid indicators were measured at various time points. Porcine neutrophil and macrophage apoptosis increased in a concentration and time dependent manner in the presence of tylvalosin. Levels of necrosis and reactive oxygen species remained unaltered. Regarding potential pro-resolution benefits, tylvalosin increased release of Lipoxin A_4_ and Resolvin D1 and inhibited pro-inflammatory Leukotriene B4. Phospholipase C from neutrophils was increased in the presence of tylvalosin while IL-8 and IL-1 alpha were inhibited. From investigation of the pharmacokinetics/pharmacodynamics of tilmicosin in sheep and cattle, Modric et al. reported on a number of complete cell count parameters [68]. The number of segmented white blood cells (WBCs) and lymphocytes decreased over time, however, the total number of WBCs, red blood cells, hemoglobin level, hematocrit, platelets and spun hematocrit increased.

Figure 4 is a schematic diagram of a cow, various immunological cells and medication and the effect points of the different impacts of macrolides based on antibacterial and non-antibacterial properties. Readers are referred to the review article for the complete list of primary research publications for each pathway or effect.

In both humans and dogs, gingival overgrowth may occur in patients receiving cyclosporine treatment but it remains uncommon. The pathogenesis appears multifunctional and possibly related to poor oral hygiene, high plaque and gingivitis, bacterial polysaccharides being present and changed to calcium ion inflex [200,201] and anticonvulsants, immunosuppressants and calcium change blockers [202]. Cyclosporine specifically increases fibroblast production of collagen and protein giving rise to extracellular collagen and matrix formation and decrease collagenase activity while increasing interleukin 6 and TGF-beta, thereby causing fibroblast proliferation [203,204]. Cyclosporines may indirectly reduce gamma-interferons which is an inhibitor of collagen synthesis [205,206].

Clementini and colleagues performed a systematic review of the efficacy of azithromycin therapy in patients with gingival overgrowth induced by cyclosporine A therapy [207]. Only 5 articles published between 1966–2008 met the review inclusion criteria of randomized control trials and a quantitative meta-analysis was not possible. Regardless, the authors summarized that a 5-day course of azithromycin with scaling and root planing reduces the degree of gingival overgrowth. In comparison, a 7-day course of metronidazole was only effective on concomitant bacterial over-infection. They concluded that further investigations were needed and the current literature was sparse. Readers are referred to the review article for the complete list of primary research publications for each pathway or effect.

In dogs, gingival overgrowth may also be related to a genetic predisposition affecting collies, boxers, great danes, Doberman pinschers and dalmatians [202,208]. Rosenberg and colleagues evaluated azithromycin in both systemic and toothpaste formulations for treatment of cyclosporine-associated gingival overgrowth in dogs. The study evaluated 36 client-owned dogs that were randomly assigned to 1 of 4 groups: (1) azithromycin capsules (10 mg/kg daily), (2) azithromycin toothpaste (8.5% with brushing once daily), (3) placebo capsule and (4) placebo toothpaste. Treatments were for 4 weeks and gingival sulcus depth, tooth length and subjective global scores were recorded at time 0 and at 2, 4 and 8 weeks post treatment. A significant decrease in gingival sulcus depth was seen in the azithromycin capsule group at 8 weeks and at weeks 2, 4 and 8 for the azithromycin toothpaste group. The mean decrease in gingival sulcus depth was greater in the azithromycin group (*p* = 0.0356). One dog in the azithromycin group had complete resolution. Gastrointestinal adverse events were more common in the azithromycin capsule group. The authors concluded that further studies were warranted and that earlier initiation of therapy, prior to severe changes, are likely beneficial.

Diesel and Morello reported on the medical management of cyclosporine-induced gingival overgrowth in 6 dogs using oral azithromycin [209]. Azithromycin therapy was 6.6–10.8 mg/kg once daily for 4–14 weeks and azithromycin induced adverse events (with long term therapy) did not occur in any dogs. The animals were receiving cyclosporine therapy for various dermatological conditions and the dogs ranged in age from 2.5–9 years. Cyclosporine therapy at the time of gingival overgrowth diagnosis ranged from 1.5–24 months. The weeks to clinical resolution ranged from 4–14 weeks and relapse occurred in only one dog with remission induced within 8 weeks of retreatment. The authors reported significant clinical improvement within the first 4 weeks of therapy and complete clinical resolution by 4–14 weeks depending on the dog. The authors also point out that in some dogs, the cyclosporine dose was decreased with concurrent azithromycin therapy but in 4 dogs the dose remained unchanged. They encouraged further investigations with different azithromycin formulations including oral tablets, compounded capsules and toothpaste. Readers are referred to the review article for the complete list of primary research publications for each pathway or effect.

As to the mechanism associated with azithromycin use for this condition, it is largely unknown but might be related to immunomodulatory properties of the drug. For example, azithromycin may block cyclosporine-induced cell proliferation and collagen synthesis while simultaneously activating metalloproeinase-2 in gingival fibroblasts and reducing hyperplastic lesions [210]. Additionally, macrolides may contribute to down regulation of transforming growth factors which could contribute to decreased overgrowth [211].

## 11. Specific Macrolide Anti-Inflammatory or Immunomodulatory Effects

Macrolides accumulate in cells and tissues including white blood cells, fibroblasts and epithelial cells [212].

Altenburg et al. summarized the cellular and non-cellular effects of macrolides and identified 17 factors including [213]:(1)Attenuations of biofilm function(2)Suppression of bacterial quorum sensing(3)Decrease in bacterial adherence(4)Loss of flagellar mobility(5)Reduced production of bacterial pathogens(6)Consolidation of epithelial tight junctions(7)Increasing ciliary beat frequency(8)Reduction of sputum quantity(9)Diminished sputum viscosity(10)Inhibition of synthesis of proinflammatory agents by bacteria, eosinophils, neutrophils, epithelial cells(11)Reduction of neutrophil chemotaxis(12)Stimulation of neutrophil degranulation(13)Acceleration of neutrophil apoptosis(14)Down regulation of adhesion molecules(15)Stimulation of phagocytosis by alveolar macrophages(16)Reduction in T-cell numbers and T-cell migration(17)Modulation of dendritic cell function

These authors suggested macrolide compounds devoid of anti-infective properties could avoid the concern related to antimicrobial resistance but provide immunmodulating properties as part of a therapeutic regimen.

Sauer and colleagues commented on antibiotics as immunomodulators for acute respiratory distress syndrome (ARDS) in humans—a condition with a mortality rate ~40% [214]. While their review summarized a number of different antibiotic classes, specifically focusing on macrolides, the authors reported several pathways that are affected by macrolides. They include:(1)intracellular killing,(2)phagocytosis,(3)macrophage maturation,(4)chemokine release,(5)inflammasome nuclear factor-ĸB activation,(6)pro-inflammatory cytokine release,(7)neutrophil efflux,(8)apoptosis,(9)oxidative damage,(10)fibroproliferation(11)apoptosis of alveolar epithelial cells

They further describe that in a healthy alveolus in humans the epithelium consists of a continuous monolayer of type I and II alveolar epithelial cells and a surfactant layer. With injured alveolus, there is sloughing of the bronchial epithelium, apoptosis of alveolar epithelial cells, oedema fluid, increased permeability of the basement membrane, edema in the interstitial space and numerous pro-inflammatory cytokines and subsequently cytokine release.

Similarly, Reijnders et al. commented on immunomodulation by macrolides and the therapeutic potential for critical care [215]. In a comprehensive schematic, they identified numerous pathways where macrolides have immunomodulatory effects during lung infection. These included:(1)biofilm and quorum sensing,(2)release of toxins and other pathogen-associated molecular patterns (PAMP),(3)phagocytosis,(4)Toll-like receptor expression and signaling,(5)intracellular killing,(6)chemokine release,(7)NETosis—neutrophil extracellular trap release,(8)chemotaxis,(9)antimicrobial peptides,(10)pro-inflammatory cytokine release,(11)tolerogenic monocyte differentiation,(12)monocyte and lymphocyte apoptosis,(13)neutrophil apoptosis,(14)release of damage-associated molecular pathogens (DAMPS),(15)efferocytosis and(16)mucous production.

## 12. Macrolides in Combination with Other Antimicrobials

### 12.1. Human Medicine

Macrolides have been recommended for use in combination with other antimicrobial agents in human medicine. Konig and colleagues commented that the role of beta lactam/macrolide combination therapy for moderately severe community acquired pneumoniae (CAP) is a “matter of debate”, however macrolides are known to extend the spectrum to cover atypical pathogens and “attenuate pulmonary inflammation” [216]. In the current therapeutic guidelines for standard empiric therapy for severe CAP both a beta lactam/fluoroquinolone combination (strong recommendation with low quality of evidence) and beta lactam/macrolide combination (strong recommendation with moderate quality of evidence) are considered acceptable but with stronger evidence in support of the beta lactam/macrolide combination [217]. Standard regimen for a non-severe inpatient CAP can be a beta lactam/macrolide combination (strong recommendation with high quality of evidence) or a respiratory fluoroquinolone (strong recommendation with high quality evidence) with a third option being a beta-lactam/doxycycline combination (conditional recommendation with low quality of evidence). Ito et al. reported azithromycin in combination with a beta-lactam agent reduced 30 day mortality in patients with severe CAP that met specific clinical criteria [218]. Antibiotic recommendations vary based on antimicrobial resistance prevalence and/or risks for specific pathogens such as methicillin-resistant *S. aureus* or *P. aeruginosa.* Bonne and Kadri summarized data on various antimicrobials and clindamycin therapy in necrotizing soft tissue infections [219] with clindamycin potentially having a number of advantages over beta-lactam therapy including an effect independent of inoculum size or infection stage and mitigate severity of shock for decreasing toxin production [220]. Hodille and colleagues [221] reported that clindamycin suppresses virulence expressions in inducible clindamycin-resistant *S. aureus* strains and indicated its usage should be considered within the treatment of toxin related infections where inducible clindamycin resistance is a concern.

### 12.2. Antibiotic Combinations in Veterinary Medicine

Antibiotic combinations, including macrolides, have been used in veterinary medicine. Prohl et al. reported on enrofloxacin and macrolides alone or in combination with rifampicin in a bovine model of *Chlamydia psittaci* infection [72]. *C. psittaci* can cause respiratory disease in human and in cattle. In this study, 50 animals were inoculated intrabronchially at age 6–8 weeks and treatment was started 30 h after inoculation with the following treatment groups: 7 untreated controls, rifampicin alone, enrofloxacin alone, enrofloxacin plus rifampicin, azithromycin alone, azithromycin and rifampicin, erythromycin alone, erythromycin plus rifampicin. All animals were successfully infected and adequate antibiotic levels were detected in plasma and tissues of infected treated animals. *C. psittaci* was isolated more frequently from untreated animals followed by rifampicin treatment at 3 days post inoculation. At 9 and 14 days post inoculation, organism was only recovered from 1 animal treated with erythromycin and rifampicin and from untreated controls (4 and 9 days post inoculation and 1 at 14 days post inoculation) but not from any other treatment group. Blood leukocytes increased 1.7 fold within 2 days after inoculation in all animals but dropped to 80% of baseline one day later and by day 14 post inoculation returned to baseline. The only statistically significant difference was seen for enrofloxacin/rifampicin treated animals on day 3 post inoculation (*p* = 0.009). Rifampicin treated animals (2 days post inoculation) had higher numbers of total blood leukocytes and neutrophilic granulocytes than did other treatment groups but the observation was not statistically significant from other treatment groups. The measurement was taken prior to rifampicin treatment and as such seems unrelated to the drug. Despite the percentage of lymphocytes decreasing by ½ within 2 days post inoculation, however, these values were at baseline by 5 days post inoculation and remained there until the end of the study. Macrolides and fluoroquinolones alone or in combination with rifampicin prohibited proliferation of *C. psittaci* in a living host, however, the study could not identify any impact on clinical illness, severity of local and systemic inflammation or the number of genetic copies detectable in tissues at 14 days post inoculation. Part of the explanation may be related to the observations that infection with this organism in humans and animals may be self-limiting and host immune defenses are adequate to deal with infection [72,222,223].

Yu et al. investigated combination antibiotic strategies against 11 multi-drug resistant *Streptococcus suis* strains [224]. In this report, they used a checkerboard assay to determine synergistic or antagonistic activity and time kill assays to measure the log_10_ reduction in viable cells. The drugs investigated included ceftiofur, tiamulin, ampicillin, apramycin, chloramphenicol, enrofloxacin, florfenicol, spectinomycin, tetracycline, trimethoprim-sulfamethoxazole, clindamycin and erythromycin. By checkerboard assay, ampicillin plus apramycin and tiamulin plus spectinomycin showed synergism against the *S. suis* strains.

Sweeney and colleagues investigated in vitro tulathromycin and ceftiofur combined with other antimicrobial agents (ampicillin, danofloxacin, enrofloxacin, florfenicol, penicillin G, tetracycline and tilmicosin) against *P. multocida* and *M. haemolytica* bovine isolates [225]. Some 458 organism-drug combinations were investigated and of these, 160 tulathromycin and 209 ceftiofur combinations with 8 other antimicrobial agents showed an indifferent effect. Only one combination showed antagonism (*P. multocida*—ceftiofur and florfenicol). The authors concluded that despite predominant results showing indifference for the antibiotic combinations tested, the results should not be interpreted as supporting concurrent antimicrobial administration for BRD treatment, however, in vivo clinical correlation was needed.

Morbidity and mortality are the usual measurements when comparing antimicrobial therapy in infectious diseases. Booker and colleagues [226] investigated concomitant therapy for “arrival fever” in ultra-high risk calves for developing undifferentiated fever/bovine respiratory disease. Treatment groups consisted of 563 animals from 4 sites receiving a subcutaneous injection of tulathromycin (2.5 mg/kg BW) and a subcutaneous injection of ceftiofur (6.5 mg/kg BW) or 563 animals from 4 sites receiving a subcutaneous injection of 40 mg florfenicol plus 2.0 mg flunixin meglumine/kg BW. Animals were followed from first treatment to up to 120 days on trial. Trial results showed the first arrival fever was lower in animals receiving tulathromycin/ceftiofur than those receiving florfenicol/flunixin with an absolute difference of 15.63% (*p* < 0.001), however, there were some differences between sites and experimental group. Other statistically significant differences (tulathromycin/ceftiofur vs. florfenicol/flunixim) between the treatment regimes includes wastage (0.89% vs. 1.78%, *p* = 0.014), overall mortality (8.7% vs. 17.05%, *p* < 0.001), BRD mortality (5.86% vs. 12.79%, *p* < 0.001) and other cause mortality (2.13% vs. 3.2%, *p* < 0.001). From data to feedlot exit from the study, the following statistically significant differences were seen between tulathromycin/ceftiofur versus florfenicol/flunixin treated animals: first arrival fever relapse treatment (24.51% vs. 39.79, *p* < 0.001), second arrival relapse fever treatment (37.68% vs. 44.2%, *p* = 0.011), chronicity (2.66% vs. 3.02%, *p* < 0.001), overall mortality (9.95% vs. 19.54%, *p* < 0.001), BRD mortality (6.22% vs. 12.97%, *p* < 0.001), metabolic mortality (0.36% vs. 1.07%, *p* < 0.001) and other cause mortality (2.84% vs. 4.26%, *p* < 0.001). The authors identified a significant interaction existed between site and experimental group for the 1st arrival fever relapse treatment. Regardless, the overall observations in this report with statistically significant morbidity/mortality differs between regimes in favour of tulathromycin/ceftiofur suggest combination therapy (with a macrolide) has benefits that may extend beyond antibacterial properties of some drugs.

In a study from Poland, Dudek et al. [227] compared enrofloxacin (5 mg/kg BW for 3 days) alone versus enrofloxacin plus flunixin meglumine (2.2 mg/kg BW for 3 days) and enrofloxacin plus flunixin meglumine plus pegbovigrastim (immunostimulant) (2.8 mg/calf once and then repeated after 7 days) for therapy in 28 days to 7 week old healthy calves infected with *Mycoplasma bovis* pneumonia. Interestingly, enrofloxacin alone stimulated a strong immune response and reduced the clinical manifestation and lung lesions while the combination therapy appeared ineffective.

Coetzee and co-investigators [228] from Iowa, USA reported on the association between antimicrobial drug classes for treatment of bovine respiratory disease and subsequent related resistance based on pathogen recovered in a veterinary diagnostic laboratory. Antimicrobials investigated included beta-lactams, fluoroquinolones, phenols, tetracycline and macrolides. An increase in the number of treatment is associated with a greater probability of resistance to at least one drug. Interestingly, these authors found treatment strategies using a bacteriostatic drug first followed by retreatment with a bactericidal drug was associated with a higher frequency of resistant pathogen isolation. First treatment with tulathromycin followed by ceftiofur was associated with highest probability of resistance with *M. haemolytica*. The authors acknowledge the data could be biased by isolates from non-responding cases of BRD as responding cases are excluded from data analysis. The potential use of concurrent antimicrobials in feed and the documented information on antimicrobial combination being limited from some combinations and more abundant for others may confound interpretation.

The use of an antimicrobial agent and anti-inflammatory agents for treating bovine respiratory disease is not a new concept. Espinasse and colleagues [229] compared oxytetracycline and chloramphenicol with or without corticosteroids for treating BRD. Animals with BRD were treated for 5 consecutive days by intramuscular injection at a rate of 20 mg/kg BW. The major pathogens were *P. multocida* and *M. haemolytica*. The antimicrobial regimen plus steroid accelerated recovery without therapeutic failure, relapse or recurrence and lead the authors to recommend continued use of steroids for BRD treatment.

Tulathromycin in combination with ketoprofen (non-steroidal anti-inflammatory) has been investigated as a single administration treatment for naturally occurring BRD [54]. A total of 140 animals each were randomized to receive either tulathromycin (2.5 mg/kg) or tulathromycin (2.5 mg/kg) plus ketoprofen (3 mg/kg) subcutaneously. Treatment success rates were 95.0% and 94.2%, respectively with relapse rates of 4.0% and 3.8%. Super pyrexia control was seen in tulathromycin/ketoprofen treated animals (*p* ≤ 0.0072) with the maximal difference seen at 6 h post-treatment (reduced by 1.33 °C). Temperature decline (for animals with fever > 40 °C) was faster in animals receiving the combination therapy than tulathromycin done in the first 24 h of treatment but not thereafter in the 2–4 days evaluation period.

## 13. Conclusions

Macrolides have been in clinical use for more than 60 years and for a wide range of clinical conditions where they have impacted morbidity and mortality. Aside from their antimicrobial properties, macrolide compounds have been extensively investigated for non-antimicrobial properties and the abundance of data available clearly show a wide impact on cellular, signaling and cytokine activities during inflammation—both in human and veterinary medicine. The non-antimicrobial effects of macrolides have been demonstrated in both acute and chronic infections and in chronic conditions where inflammation contributes to functional decline or deterioration of, e.g., Pulmonary function. The lack of consistency in observations between various studies have been related to numerous variables including intrinsic differences between macrolide compounds, lack of standardized assay between studies, different parameters being measured and even intrinsic differences between humans or animals enrolled in studies. Regardless there is sufficient evidence showing clear immunomodulatory effects of macrolide compounds. Of course, the use of antimicrobials without clear clinical benefits from the antimicrobial activity is always a concern for antimicrobial resistance—particularly with long term use. Llor and Bjerrum identified several risks associated with over use of antibiotics including increased antimicrobial resistance, increase in more severe disease, increase in length of disease, increase in risk of complications, increased mortality, increased health care costs, increased adverse events (some life threatening), increased re-attendance from infectious diseases and increased medicalization of self-limiting infectious conditions [230]. As such, antimicrobial use needs to be carefully considered.

Bosnar and colleagues reported on novel anti-inflammatory macrolides without antimicrobial activity [231]. They argued that antimicrobial and non-antimicrobial activity of macrolides are independent and separatable with the potential for development of anti-inflammatory therapeutic agents. In an editorial, Brusselle asked “are the antimicrobial properties of macrolides required for their therapeutic efficacy in chronic neutrophilic airway disease?” From literature reviewed, it appeared immunomodulatory properties were more important for the effects seen. As such, it is another example of the non-antimicrobial benefit of macrolides in some clinical scenarios. Clearly, the various properties of macrolides and the potential clinical benefits continue to be defined and ongoing and future investigations are warranted.

## Figures and Tables

**Figure 1 microorganisms-10-02438-f001:**
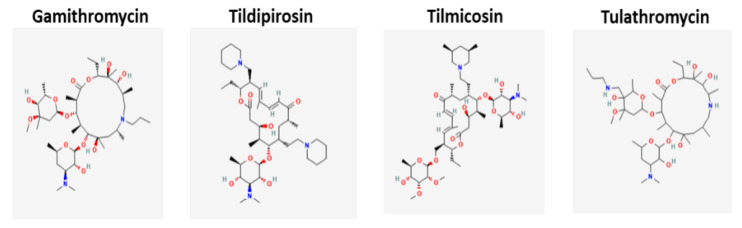
Molecular structures of 4 veterinary macrolides.

**Figure 2 microorganisms-10-02438-f002:**
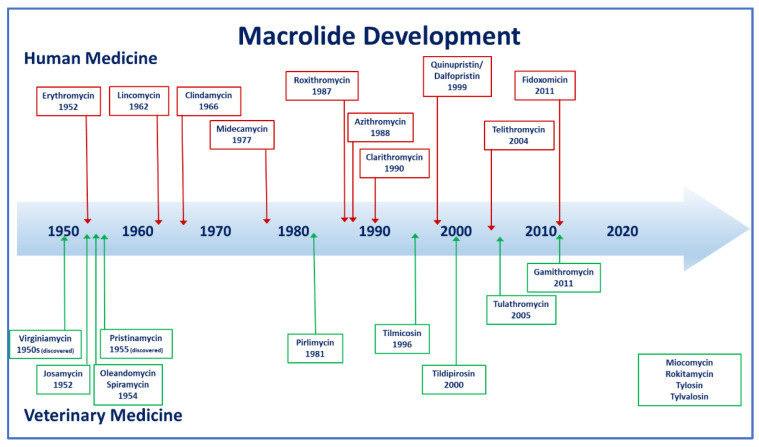
Chronology of macrolide development in human and veterinary medicine.

**Figure 3 microorganisms-10-02438-f003:**
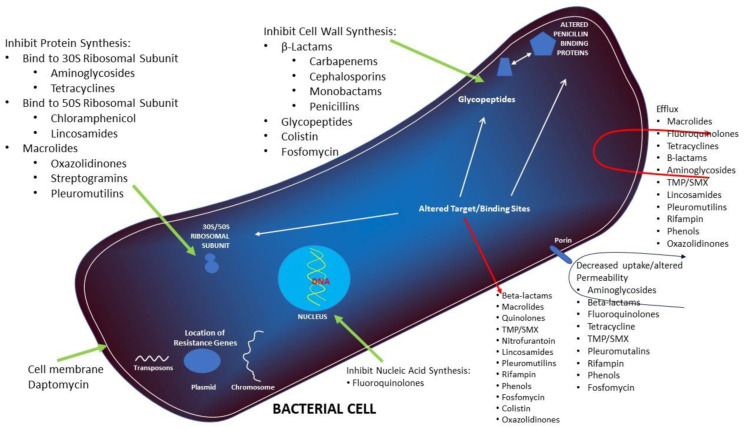
Antibiotics mechanisms of action (green arrows) and bacterial mechanisms of antibiotic resistance (red arrows). TMP/SMX = trimethoprim/sulfamethoxazole. (Reproduced with permission).

**Figure 4 microorganisms-10-02438-f004:**
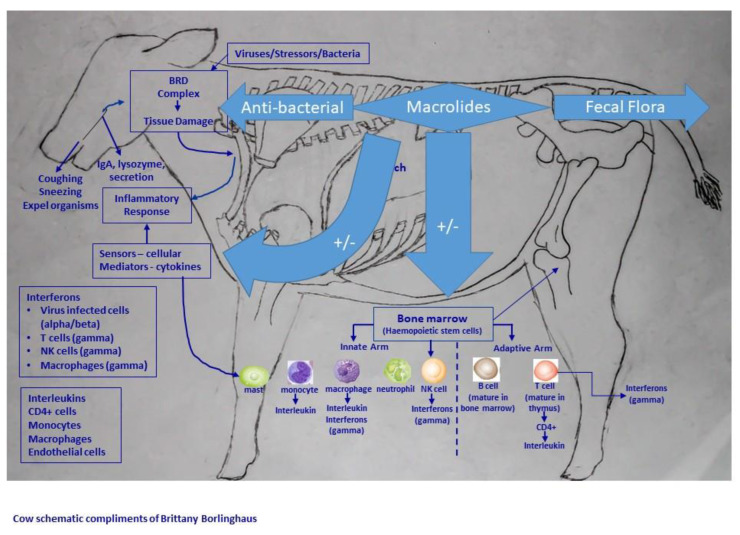
Schematic diagram of a cow summarizing various compounds of the immune system and points of macrolide interactions. (+ = stimulation, − = inhibition).

**Table 1 microorganisms-10-02438-t001:** Veterinary macrolides and clinical indicators.

Drug	Indications	Organism on Label
Pirlimycin	Mastitis in cattle [9]	*S. aureus* *S. agalactiae* *S. dysagalactiae* *S. uberis*
Tulathromycin	BRD * [10]	*M. haemolytica* *P. multocida* *H. somni* *M. bovis*
	SRD * [11]	*A. pleuropneumoniae* *H. parasuis* *P. multocida* *B. bronchiseptica*
	IBKC * [12]	*Moraxella bovis*
	Bovine Foot Rot [13]	*Fusobacterium necrophorum,* *Porphyromonas levii*
Gamithromycin	BRD [14]	*M. haemolytica* *P. multocida* *H. somni* *M. bovis*
Tildipirosin	BRD [15]	*M. haemolytica* *P. multocida* *H. somni*
Tilmicosin	BRD	*M. haemolytica* *P. multocida* *H. somni*
Tylosin	BRC	*P. multocida* *A. pyogenes*
	Foot rot/calf diphtheria	*F. necrophorum*
	Metritis	*A. pyogenes*
	Swine arthritis	*M. hyosynoviae*
	Swine pneumonia	*Pasteurella* spp.
	Swine erysipelas	*E. rhusiopathiae*
	Swine dysentery	*T. hyodysenteriae*

* BRD = bovine respiratory disease; IBKC = infectious bovine keratoconjunctivitis; SRD = sheep respiratory disease.

**Table 2 microorganisms-10-02438-t002:** Comparative in vitro activity of four veterinary macrolides against select veterinary pathogens.

	Gamithromycin			Tildipirosin			Tilmicosin			Tulathromycin		
Organism	No.	MIC_50/90_	Range	No.	MIC_50/90_	Range	No.	MIC_50/90_	Range	No.	MIC_50/90_	Range
***A. pleuropneumoniae* [41]**							67	2/4	1–8	67	1/2	0.5–8
**Swine [42]**							242	4/8	1–256	242	8/16	1–32
**Swine 2009–2012 [43]**							158	16/16	2–32	158	32/32	4–64
**Swine 2017–2019 [44]**				162	4/8	1–64	162	8/16	4–64	162	32/64	8–64
***P. multocida* (swine) [41,45]**							73	2/4	1–8	73	0.25/0.5	0.063–1
**2009–2012**							152	8/16	1–32	152	2/4	1–8
**2017–2019 [44]**				130	1/2	0.5–64	130	8/16	1–64	130	2/4	0.5–65
***S. suis* [41,45]**							59	≥8/≥8	≥8	59	≥8/≥8	≥8
**2018–2022 [46]**							246	128/128	4–128	246	64/64	1–64
**Swine 2009–2012**							152	>64/>64	0.5–>64	151	>64/>64	1–>64
***M. haemolytica* [45]**	89	1/1	0.5–>32									
**2014–2018 [47]**	55	0.5/1	1–2	55	1/1	1–2	55	4/8	4–16	55	8/16	1–16
**2019 [47]**	33	1/2	1–16	33	1/1	1–4	33	4/8	2–32	33	8/8	8–128
**2014–2015 [48]**							251	16/64	4–64	251	16/64	1–64
**2009 [49]**							304	8/64	1–>64	304	8/32	0.5–>64
**2009–2012**	149	0.5/1	0.12–1				149	8/16	1–64	149	4/8	1–128
***P. multocida* (bovine)**	79	0.5/1	0.12–>32									
**2014–2018 [47]**	75	1/8	1–16	75	1/8	1–16	75	4/8	4–64	75	1/8	0.5–128
**2019 [47]**	65	1/16	1–16	65	1/2	1–16	65	8/32	2–32	65	8/64	8–64
**2014–2015 [48]**							118	16/64	4–64	118	4/64	1–64
**2009 [49]**							328	8/>64	0.5–>64	328	2/16	0.5–>64
**2009–2012**	134	0.25	0.06–2									
***H. somni* [50]**	32	0.5/0.5	0.25–1									
**2014–2015 [48]**							80	4/≥64	4–≥64	80	8/≥64	1–≥64
**2009 [49]**							174	8/64	2–>64	174	8/>64	1–>64
**2009–2012**	66	0.5/0.5	0.25–1				66	4/8	0.5–16	66	4/8	0.12–8
** *R. equi* **												
**Macrolide S [51]**		1/1	0.5–1									
**Macrolide R [51]**	30	64/128	32–128									
** *S. zooepidemicus* **	22	0.06/0.125	0.03–0.125									
***B. bronchiseptica* (Swine)**												
**2009–2012**							118	32/32	8–64	118	8/8	1–32
**2017–2019 [44]**				29	4/8		29	32/64		29	8/8	
** *M. bovis* **												
**2014–2015 [48]**							234	64/64	4–64	234	64/64	1–64
** *T. pyogenes* **												
**2014–2015 [48]**							94	64/64	4–≥64	94	64/64	1–≥64
***H. parasuis* (Swine)**												
**2009–2012**							68	1/2	0.12–4	68	½	0.06–4

**Table 3 microorganisms-10-02438-t003:** Immunomodulatory markers affected by macrolides.

Marker	Function	Macrolide Impact
Total cell count	Total # of cells in a given area.	Significant decrease or no change—clarithromycin. Significant decrease—erythromycin.
Leukocyte count	# of white blood cells in blood.	Significant decrease—azithromycin. Leukocyte viability reduced (age dependent)—bovine—tylosin, tilmicosin, roxithromycin [84].
Neutrophil count	# of neutrophils in blood.	Significant decrease—azithromycin, clarithromycin, erythromycin, roxithromycin. No change—azithromycin, clarithromycin. Non-significant change—clarithromycin. Airway neutrophilia reduced in 4/6 horses. Non-significant—azithromycin.
Neutrophil oxidative burst	Reactive oxygen intermediates (ROI) during phagocytosis—antimicrobial.	Non-significant decrease—clarithromycin. Significant decrease—roxithromycin, erythromycin, azithromycin. Significant increase—azithromycin.
Neutrophil chemotaxis	Migration of neutrophils.	Non-significant decrease clarithromycin. Significant increase—erythromycin. Significant decrease—azithromycin, erythromycin. Reduced chemotactic activity—bovine leukocytes roxithromycin, tylosin, tilmicosin. Inhibition by erythromycin, roxithromycin, azithromycin [85,86].
Neutrophil lactoferrin	Iron binding protein Released from activated neutrophils. Antimicrobial and anti-inflammatory properties.	Significant decrease—azithromycin.
Neutrophil elastase	Proteolytic enzyme required for neutrophil function. Inflammatory response to tissue injury.	Significant decrease—clarithromycin, roxithromycin, erythromycin. No change clarithromycin—azithromycin. Non-significant decrease—azithromycin, clarithromycin. Inhibition by erythromycin [86].
Mucus	Traps allergens, dust particles, debris from entering lungs.	No decrease in mucus accumulation—azithromycin [87].
Neutrophil apoptosis	Controlled and programmed cell death.	No effect—horses—azithromycin. Significant reduction of bovine leukocytes [84]. Strongest effect—tilmicosin, tylosin, roxithromycin. Tilmicosin induced apoptosis (significant) in bovine neutrophils [88].
Macrophage count	# of macrophages. Frontline defense. Ingest and kill pathogens	Significant decrease—clarithromycin. Significant increase—clarithromycin. Non-significant increase—clarithromycin.
Eosinophil count	# of eosinophils Traps and kills cells. Isolates and control immune response at site of infection. Modulates inflammatory response.	No change—azithromycin. Significant decrease—clarithromycin, roxithromycin.
Eosinophil cationic protein	A ribonuclease. Regulates mucosal and immune cells. May directly act against helminth, bacterial and viral infections. Released during degranulation of eosinophils.	Significant decrease—clarithromycin, roxithromycin.
Thrombocyte count	# of thrombocytes. Help form blood clots to stop bleeding and help wounds heal.	Significant decrease—azithromycin. No change—azithromycin.
IL-1beta	A cytokine. Mediator of inflammatory response Leukocyte pyrogen, leukocytic endogenous mediator, mononuclear lymphocyte activating factor.	Significant decrease—azithromycin, clarithromycin, erythromycin, roxithromycin. No change—azithromycin. Significant decrease—horses—azithromycin. Reduced production—erythromycin.
IL-2	A cytokine. Regulates white blood cells (leukocytes + lymphocytes). From activated CD4+ and CD8+ T cells.	Significant decrease—erythromycin. Non-significant decrease—clarithromycin.
IL-4	A cytokine. Stimulation of activated B/T cell proliferation. Differentiation of B cells into plasma cells.	Significant decrease—azithromycin, clarithromycin. Significant decrease—erythromycin.
IL-5	A cytokine—T cell derived. Controls production, activation and localization of eosinophils. Stimulates B cell growth.	Increases immunoglobulin secretion (IgA). Significant decrease—azithromycin, clarithromycin. Significant increase—erythromycin.
IL-6	A cytokine—produced from fibroblasts, keratinocytes, mesangial cells, vascular and endothelial cells, most cells, macrophages, dendritic cells, T cells and B cells. Stimulating acute phase protein synthesis. Stimulates production of neutrophils. Supports growth of B cells. An antagonistic to regulating T cells.	Significant decrease—clarithromycin, azithromycin. No change—azithromycin. Non-significant decrease—clarithromycin. Reduced production—erythromycin, roxithromycin, clarithromycin.
IL-8	Neutrophil chemotactic factor. Stimulate phagocytosis. Key role in recruitment of neutrophils/immune cells to infection site. Released by macrophages, epithelial cells, airway smooth muscle cells, endothelial cells.	Significant decrease—azithromycin, clarithromycin, erythromycin, roxithromycin. No change—clarithromycin. Non-significant decrease—azithromycin, clarithromycin. Reduced production—erythromycin, clarithromycin [89].
IL-12	A cytokine. Produced by dentritic cells, macrophages, neutrophils, human B-lymphoblastoid cells. Differentiates naïve T cells into Th cells.	Significant decrease—clarithromycin. Significantly reduced mRNA expression of IL-8 in horses—azithromycin.
IL-17	A cytokine. T cell activation. Neutrophil mobilization/activation. Biomarker for sepsis. Produced by Thelper 17 cells.	Significant decrease—azithromycin, clarithromycin. IL-17A—horses—unchanged—azithromycin.
TNF- alpha	Tumour necrosis factor—inflammatory cytokine. Produced by macrophages/monocytes. Diverse range of signaling—leads to necrosis or apoptosis. Regulates immune cells. Endogenous pyrogen—induces fever, apoptatic cells death, cachexia.	Non-significant decrease—clarithromycin. Non-significant increase—azithromycin. Unchanged—horses—azithromycin. Reduction production—erythromycin, clarithromycin, roxithromycin.
Lymphocytes	Produced from bone marrow. B lymphocytes produce antibodies. T lymphocytes—kill tumour cells and regulate immune responses.	Bovine lymphocytes affected less by macrolides (age variability) [84].
IFN-gamma	Cytokine. Critical for innate/adaptive immunity. Primary activator of macrophages. Stimulates natural killer cells, neutrophils.	Significant decrease—azithromycin, erythromycin. No change—azithromycin. Non-significant decrease—clarithromycin.
TGF-beta	Transforming growth factor. Cytokine. Important roles: wound healing, angiogenesis, immune-regulation. Secreted by macrophages, T cells, monocytes.	Significant increase—azithromycin. Significant decrease—roxithromycin. Non-significant decrease—clarithromycin.
Monocytes	Phagocytic while blood cell. Inflammatory and anti-inflammatory processes during immune response.	Decreased viability—bovine monocytes—age variability—not significant.
GM-CSF	Granulocyte—macrophage—colony stimulating factor. Monomeric glycoprotein—function as a cytokine. White blood cells growth factor. Stimulates stem cells to produce neutrophils/ basophils. Immune-modulator.	No change—azithromycin.
VEGF	Vascular endothelial growth factor. Stimulates formation of blood vessels. Restores oxygen supply to tissues.	Significant decrease—azithromycin, roxithromycin.
Matrix metalloproteinase-9	Regulates tissue remodeling. Degrades extracellular matrix proteins. Activator cytokines/chemokines for regulating tissue remodeling. Important in wound healing.	Significant decrease—azithromycin, clarithromycin, roxithromycin. Non-significant decrease—clarithromycin.
E-selectin	Inducibly expressed by cytokines on inflamed endothelium. Selection cells adhesion molecule. Expressed on endothelial cells activated by cytokines.	Significant decrease—azithromycin. Non-significant decrease—azithromycin.
C-reactive protein	Made by liver. Released in blood stream in response to inflammation. Release is triggered by inflammatory cytokines. Innate immune system. Surveillance molecule for detecting altered self or pathogens. Activation of humoral, adaptive immune systems. Opsonin for various pathogens.	Significant decrease—azithromycin. Non-significant decrease—clarithromycin.
Serum amyloid A	Cytokine like protein. Cell–cell communication. Produced by liver. Induces synthesis of cytokines. Chemotactic for neutrophils/mast cells.	Significant decrease—azithromycin.
ICAM-1 (C054) (Intracellular adhesion molecule)	Endothelial and leukocyte-associated transmembrane protein. Co-stimulatory molecule-presenting cells. Activates MHC class II restricted T cells.	Reduced gene expression/production of ICAM-1—erythromycin, clarithromycin, roxithromycin [85].

# = number. Tissues/fluids analyzed: nasal secretions/mucosa [90,91,92,93,94,95,96,97,98], conjunctiva [99], gingival fluid [100,101], sputum [102,103,104,105,106,107,108,109,110,111,112,113,114], BAL [86,115,116,117,118,119,120,121,122,123,124,125], airway tissue [116], blood [104,107,109,113,125,126,127,128,129,130], PMNL [131].

## Data Availability

The data that support the findings of this study are available from the corresponding author upon reasonable request.

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
