# Peer review of "Immunomodulatory Effects of Macrolides Considering Evidence from Human and Veterinary Medicine"

_microorganisms, 2022, doi:10.3390/microorganisms10122438_

Round 1

Reviewer 1 Report

The scientific article is well written. More discussion on the use of macrolides in pets could be useful. Such as the use of azithromycin in cyclosporine-induced gingival hyperplasia in dogs, with immunomodulating activity.

Author Response

Note to Editor:  I have used our original copy submitted as the new version you wanted us to use had cancelled the endnote program and I could not add additional references as they would be listed as new references starting at #1.

Reviewer 2 Report

In this article the author provides an up-to-date review of the current literature regarding the antimicrobial and non-antimicrobial properties of macrolides in human and veterinary medicine. While the review is well written, there are some formatting issues that should be addressed, in addition to the comments below.

Major comments:

Line 98-115 – this paragraph is confusing to read. The MIC vales are already stated within the Table above, so it might be more useful to describe this and just refer to the table rather than repeating the information contained within the table in text.

For the section on mechanisms of resistance, a schematic/figure illustrating this would be beneficial.

Table 3: Is there references to support the effect of macrolides on each of the functions listed. Again, a schematic or figure would be useful to complement the table.

Figure 3: Figure 3 is unclear, and of poor quality. Please increase image quality to make more clear and detailed for the reader.

Line 751-776: Please reformat. Rather than listing 1-17, a figure would be useful. Please include references for each of the stated effect.

Line 778-787: As above, for each pathway that the authors are claiming that macrolides impact, please provide original reference rather than referencing those outlines in another review.

Line 799-814 – as per two comments above.

Section 12. Macrolides in combination with other antimicrobials -this section only discusses human medicine. What about veterinary medicine? This section should also be included here.

Conclusion: The authors highlight the potential of non-antimicrobial effects of macrolides. While the authors highlight the impact this could potentially have in terms of antimicrobial resistance, a separate expanded section should be included on the potential negative effects on antimicrobials, in terms of antimicrobial resistance. Moreover, antibiotics have negative effects on the microbiome, this is not mentioned within the review and is a particularly important implication of antibiotic use. A separate section should be included within the body of the review making reference to this.

Minor comments

Table 2 is a little confusing. Can this be reformatted to make for clear for the reader.

Abstract: the last sentence is a little confusing.

Throughout the review article the effects of macrolides are listed for example 1-17 on page 21-22. Reformatting of these sections in a table including the original article reference would support each statement would be useful and clearer for the reader.

Please ensure each statement is references, particularly when referring to the effects of macrolides.

Author Response

Note to Editor:  I used the original submission copy as the new file would not allow me to add more references to be included in the article.  When adding new references with revisions they started counting the references with #1 again and there were 2 sets of references.
